# Fishers for Free? Approximating the Fisher Information Matrix by Recycling the Squared Gradient Accumulator

**YuXin Li** [1]  **Felix Dangel** [1]  **Derek Tam** [1]  **Colin Raffel** [1]

## Abstract

The diagonal of a model's Fisher Information Matrix (the "Fisher diagonal") has frequently been used as a way to measure parameter sensitivity. Typically, the Fisher diagonal is estimated via squared sampled gradients of the model's likelihood with respect to its parameters, averaged over a few hundred or thousand examples – a process which incurs nontrivial computational costs. At the same time, adaptive gradient methods like the ubiquitous Adam optimizer compute a moving average of the squared gradient over the course of training. This paper therefore explores whether an approximation of the Fisher diagonal can be obtained "for free" by recycling the squared gradient accumulator that has already been computed over the course of training. Through a comprehensive set of experiments covering five applications of the Fisher diagonal, we demonstrate that the "Squisher" (**Squ**ared gradient accumulator as an approximation of the F**isher**) consistently performs similarly to the Fisher diagonal while outperforming baseline methods. Additionally, we clarify the exact differences between the Squisher and the Fisher diagonal and provide empirical quantification of their respective impact.

## 1. Introduction

The Fisher Information Matrix (FIM, Fisher, 1922) is a fundamental concept in statistics, capturing how much information an observable random variable carries about an unknown parameter. In machine learning, the FIM has been widely used in optimization, particularly in Natural Gradient Descent (NGD, Amari, 1998). Unlike standard gradient-based methods, which update parameters using the Euclidean gradient, NGD leverages the geometry from the statistical manifold of likelihoods and scales updates according to the parameter space's curvature, informing us about the "steepness" or "flatness" of the objective function at a given point in the parameter space (Karakida et al., 2019). This has made the FIM a valuable tool for training neural networks, improving stability and convergence speed (Karakida & Osawa, 2020).

Beyond optimization, recent work has explored the use of the FIM's diagonal (the "Fisher diagonal") as a measure of parameter sensitivity (Ly et al., 2017), including applications in model sparsification (Theis et al., 2018), sparse training (Sung et al., 2021), task similarity measurement (Achille et al., 2019), continual learning (Kirkpatrick et al., 2017), and model merging (Matena & Raffel, 2022; Daheim et al., 2024). These applications leverage the fact that the diagonal elements of the Fisher reflect each parameter's impact on the model output, when considering parameters to be independent, providing a principled approach to understanding and modifying neural nets.

Despite its utility, computing the Fisher diagonal introduces nontrivial computations beyond those for training. While these costs are typically comparable to training on a few hundred or a few thousand examples, the Fisher diagonal requires computing, squaring, then summing per-example gradients on sampled labels. Doing so efficiently is nontrivial in most deep learning frameworks. While there exist specialized solutions (Dangel et al., 2020; Osawa et al., 2023), practitioners often resort to sequential gradient computations (i.e. with a "batch size of one"), which sacrifices parallelization opportunities. Additionally, computing the Fisher diagonal requires access to training data that might not be available – for example, trained models are frequently released without their associated training data. We suspect that these factors can hinder the adoption of methods that require computing the Fisher diagonal – for example, Fisher Merging (Matena & Raffel, 2022) is not implemented in the ubiquitous `mergekit` library (Goddard et al., 2024), as it does not support merging methods that require computing gradients or accessing external data.

At the same time, modern neural networks are typically trained using adaptive gradient methods such as Adam (Kingma & Ba, 2014) that make use of an accumulator vari-

---

[1]University of Toronto & Vector Institute. Correspondence to: YuXin Li <lyx.li@mail.utoronto.ca>.

*Proceedings of the 42nd International Conference on Machine Learning*, Vancouver, Canada. PMLR 267, 2025. Copyright 2025 by the author(s).

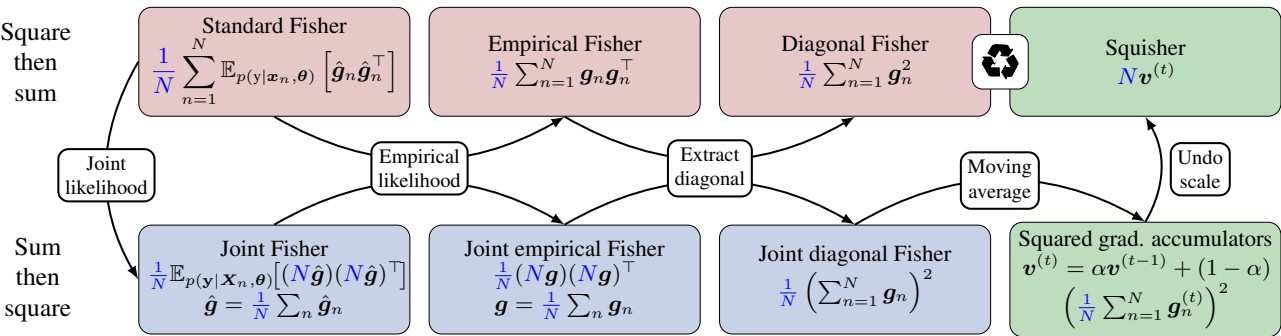

*Figure 1.* The arrows represent various approximations of the Fisher Information Matrix. The central idea of the paper is highlighted through the recycling symbol. Namely, we show that squared gradient accumulators can be used to approximate the Fisher diagonal (details in Section 2). Terms in blue stem from using a loss function with mean reduction ($1/N \mathcal{L}_{\text{sum}}$), and should be omitted when using sum reduction ($\mathcal{L}_{\text{sum}}$). The per-sample gradients $\boldsymbol{g}_n$ and "would-be" gradients $\hat{\boldsymbol{g}}_n$ are defined in Equations (3) and (5).

able to store an exponential moving average of the squared gradient over training steps. The squared gradient accumulator is at least superficially similar to the Fisher diagonal in the sense that both compute some kind of average of squared gradients. The fact that such accumulator variables are available "for free" (i.e. without requiring any additional computation) at the end of training, combined with the aforementioned inconveniences of computing the Fisher, raises a natural question for applications that use the Fisher as a notion of parameter importance:

> *Can we recycle an optimizer's squared gradient accumulator to approximate the Fisher diagonal for free?*

The superficial similarities of the Fisher diagonal and squared gradient accumulator suggest this could be trivially possible, and indeed, past work like the paper introducing Adam (Kingma & Ba, 2014) asserts without evidence that the squared gradient accumulator "is an approximation to the diagonal of the Fisher information matrix". However, a closer look reveals important subtleties. Our contribution is to rigorously assess and empirically validate these nuances:

- We discuss the non-trivial connections between the Fisher diagonal and squared gradient accumulators from adaptive optimizers, highlighting important modifications that are necessary to successfully use them in place of the Fisher (Section 2).

- We empirically validate the *Squisher* (**squ**ared gradient accumulator in place of the F**isher**) across six settings spanning five key applications. We find that the Squisher achieves comparable performance to the Fisher diagonal and consistently outperforms the baseline across all settings (Section 3). This motivates the Squisher as a practical and efficient alternative that eliminates the computational costs and inconvenience

of Fisher-based methods without sacrificing effectiveness.

- We investigate the Squisher's limitations and trade-offs, providing a deeper understanding of when and why it succeeds (Section 4).

On the whole, our work comprehensively establishes the connections between the squared gradient accumulator and the Fisher diagonal, paving the way for broader adoption of Fisher-based techniques in deep learning.

## 2. Background & Motivation

Here we provide background on, and connect, the Fisher diagonal and squared gradient accumulators (Figure 1).

### 2.1. The Fisher Information Matrix and its Variants

**Standard Fisher Information Matrix**   For simplicity, we first consider an unscaled loss of the form

$$\mathcal{L}_{\text{sum}}(\boldsymbol{\theta}) := \sum_{n=1}^{N} \ell(f(\boldsymbol{x}_n, \boldsymbol{\theta}), y_n), \tag{1}$$

where $f(\cdot, \boldsymbol{\theta})$ is a neural network with parameters $\boldsymbol{\theta}$ that processes a data point $\boldsymbol{x}_n$ into a prediction which is then scored by comparison with its label $\boldsymbol{y}_n$ using a criterion function $\ell$. Also define the per-sample gradient $\boldsymbol{g}_n := \nabla_{\boldsymbol{\theta}} \ell(f(\boldsymbol{x}_n, \boldsymbol{\theta}), y_n)$. Later, we will add back the more common scaling and consider $\mathcal{L}(\boldsymbol{\theta}) = 1/N \mathcal{L}_{\text{sum}}(\boldsymbol{\theta})$. The most common loss functions in machine learning, like square or softmax cross-entropy loss, permit the loss in Equation (1) to be interpreted as a negative log likelihood for a random variable y representing a target (Martens, 2020), and we can define a per-example likelihood

$$p(\text{y} \mid \mathbf{x}, \boldsymbol{\theta}) \propto \exp\left(-\ell(f(\mathbf{x}, \boldsymbol{\theta}), \text{y})\right). \tag{2}$$

Note the relation to the per-sample gradient when $\mathrm{y} = y_n$ and $\mathbf{x} = \boldsymbol{x}_n$

$$-\nabla_{\boldsymbol{\theta}} \log p(y_n \mid \boldsymbol{x}_n, \boldsymbol{\theta}) \overset{(2)}{=} \boldsymbol{g}_n. \tag{3}$$

The *standard Fisher Information Matrix* (FIM) is

$$\boldsymbol{F}_{\text{std}}(\boldsymbol{\theta}) := \sum_{n=1}^{N} \mathbb{E}_{\hat{y}_n \sim p(\mathrm{y}\mid\boldsymbol{x}_n, \boldsymbol{\theta})} \left[ \hat{\boldsymbol{g}}_n \hat{\boldsymbol{g}}_n^{\top} \right] \tag{4}$$

with "would-be" gradients

$$\hat{\boldsymbol{g}}_n := -\nabla_{\boldsymbol{\theta}} \log p(\hat{y}_n \mid \boldsymbol{x}_n, \boldsymbol{\theta}) = \nabla_{\boldsymbol{\theta}} \ell(f(\boldsymbol{x}_n, \boldsymbol{\theta}), \hat{y}_n) \tag{5}$$

that require drawing labels $\hat{y}_n$ from the model's likelihood.

**Empirical Fisher Information** The *empirical FIM* replaces the model's likelihood with the empirical likelihood implied by the data, $p(\mathrm{y} \mid \boldsymbol{x}_n, \boldsymbol{\theta}) \to \delta(\mathrm{y} - y_n)$. It consists of per-sample gradients which do *not* require sampling:

$$\boldsymbol{F}_{\text{std}}^{\text{emp}}(\boldsymbol{\theta}) := \sum_{n=1}^{N} \mathbb{E}_{\hat{y}_n \sim \delta(\mathrm{y}-y_n)} \left[ \hat{\boldsymbol{g}}_n \hat{\boldsymbol{g}}_n^{\top} \right] = \sum_{n=1}^{N} \boldsymbol{g}_n \boldsymbol{g}_n^{\top}. \tag{6}$$

While past work has argued against using the empirical FIM in second-order methods like NGD (Kunstner et al., 2019; Thomas et al., 2020), it nevertheless remains popular in applications where the FIM is being used as a measure of parameter importance (e.g. Matena & Raffel, 2022; Theis et al., 2018; Sung et al., 2021; Achille et al., 2019, inter alia) due its lower computational costs. We therefore primarily focus on the empirical FIM in this work.

**Diagonals** Our discussion so far was concerned with Fisher information *matrices*, which are quadratic in the number of model parameters and therefore prohibitively large in most deep learning applications. Hence, many applications only compute and use the diagonal of the FIM. Note that for a matrix that is a sum of rank one matrices, $\boldsymbol{A} = \sum_i \boldsymbol{v}_i \boldsymbol{v}_i^{\top}$, which is the case for all the Fisher flavors we previously discussed, the diagonal is $\text{diag}(\boldsymbol{A}) = \sum_i \boldsymbol{v}_i^2$ where the square is applied elementwise. This leads to simple expressions for the standard diagonal Fisher:

$$\text{diag}(\boldsymbol{F}_{\text{std}}(\boldsymbol{\theta})) = \sum_{n=1}^{N} \mathbb{E}_{\hat{y}_n \sim p(\mathrm{y}\mid\boldsymbol{x}_n, \boldsymbol{\theta})} \left[ \hat{\boldsymbol{g}}_n^2 \right]$$

and the empirical diagonal Fisher:

$$\text{diag}(\boldsymbol{F}_{\text{std}}^{\text{emp}}(\boldsymbol{\theta})) = \sum_{n=1}^{N} \boldsymbol{g}_n^2 \tag{7}$$

where the square is elementwise.

## 2.2. Gradient Accumulators

Training deep neural networks is an ill-conditioned and non-convex optimization problem (Dauphin et al., 2014; Saarinen et al., 1993). The size of typical datasets used in neural network training also necessitates stochastic optimization where different randomly sampled batches of data are used at each training step (LeCun et al., 2002; Robbins & Monro, 1951). Consequently, most optimizers used in neural network training incorporate some mechanism to provide adaptivity to noise and differences in parameter sensitivity (i.e. differences in per-parameter gradient scale). One common approach is to approximate the average squared gradient for each parameter over the training steps. This squared gradient estimate can then be used e.g. to normalize each parameter update by a smoothed estimate of the gradient magnitude.

Specifically, we focus on optimizers with a squared gradient accumulator that takes the form of an exponential moving average (EMA, Roberts, 1959) of squared gradients:[1]

$$\boldsymbol{v}^{(t)} = \alpha \boldsymbol{v}^{(t-1)} + (1 - \alpha) \left( \frac{1}{N} \sum_{n=1}^{N} \boldsymbol{g}_n^{(t)} \right)^2 \tag{8}$$

where $\alpha$ is a hyperparameter and $\boldsymbol{g}_n^{(t)}$ is the gradient vector for training example $n$ at training step $t$. This accumulator mechanism, first introduced in 2012 through concurrent developments of RMSProp (Tieleman & Hinton, 2012) and Adadelta (Zeiler, 2012), has become a fundamental technique for adapting learning rates to stabilize training dynamics. Apart from being used in the ubiquitous Adam optimizer (Kingma & Ba, 2014) and its derivatives (AdamW (Loshchilov, 2017), NAdam (Dozat, 2016), RAdam (Liu et al., 2020), AdamP (Heo et al., 2021), etc.), it is also included in FTRL (McMahan & Streeter, 2012), AMSGrad (Reddi et al., 2018), QHM (Ma & Yarats, 2019), LAMB (You et al., 2020), and many others (Schmidt et al., 2021). In this work, we focus primarily on Adam and AdamW, which represent the most commonly used optimizers in contemporary deep learning research.

## 2.3. The Squisher

**Joint Fisher** Note that both the standard and empirical FIMs in Equations (4) and (6) require per-datum (would-be) gradients to be squared then accumulated. This is in contrast to the squared gradient accumulator of Equation (8), which *first* aggregates the per-datum gradients, *then* squares them. Lin et al. (2024) proposed a new Fisher matrix which has a sum-then-square structure that is more compatible with the structure of squared gradient accumulators. Instead of modeling the label as single random variable y, they

---

[1]Here, we have already included a scaling factor of $1/N$ in the loss and gradient, as we will later use this scenario in practice.

consider a likelihood for a random vector of labels $\mathbf{y} = (y_1, y_2, \ldots, y_N)$ jointly from inputs $\mathbf{X} = (\mathbf{x}_1, \ldots, \mathbf{x}_N)$ via

$$p(\mathbf{y} \mid \mathbf{X}, \boldsymbol{\theta}) = \prod_{n=1}^{N} p(y_n \mid \mathbf{x}_n, \boldsymbol{\theta}).$$

Their *joint Fisher information matrix* is

$$\boldsymbol{F}_{\text{joint}}(\boldsymbol{\theta}) \coloneqq \mathbb{E}_{\hat{\boldsymbol{y}} \sim p(\mathbf{y} \mid \boldsymbol{X}_n, \boldsymbol{\theta})} \left[ \hat{\boldsymbol{g}} \hat{\boldsymbol{g}}^\top \right] \qquad (9)$$

with "would-be" gradients $\hat{\boldsymbol{g}} = -\nabla_{\boldsymbol{\theta}} \log p(\hat{\boldsymbol{y}} \mid \boldsymbol{X}, \boldsymbol{\theta}) = -\sum_{n=1}^{N} \nabla_{\theta} \log p(\hat{y}_n \mid \boldsymbol{x}_n, \boldsymbol{\theta}) = \sum_{n=1}^{N} \hat{\boldsymbol{g}}_n$ and $\hat{y}_n = [\hat{\boldsymbol{y}}]_n$. As for the standard case, their *joint empirical Fisher information matrix* follows by replacing the model's likelihood over the joint labels with the likelihood implied by the data, $p(\mathbf{y} \mid \boldsymbol{X}, \boldsymbol{\theta}) \to \prod_{n=1}^{N} \delta(y_n - y_n)$, which gives

$$\boldsymbol{F}_{\text{joint}}^{\text{emp}}(\boldsymbol{\theta}) \coloneqq \mathbb{E}_{\hat{y} \sim \prod_{n=1}^{N} \delta(y_n - y_n)} \left[ \hat{\boldsymbol{g}} \hat{\boldsymbol{g}}^\top \right] = \boldsymbol{g} \boldsymbol{g}^\top \qquad (10)$$

with the empirical gradient $\boldsymbol{g} = -\sum_{n=1}^{N} \nabla_{\boldsymbol{\theta}} \log p(y_n \mid \boldsymbol{x}_n, \boldsymbol{\theta}) = \sum_{n=1}^{N} \nabla_{\boldsymbol{\theta}} \ell(f(\boldsymbol{x}_n, \boldsymbol{\theta}), y_n) = \nabla_{\boldsymbol{\theta}} \mathcal{L}_{\text{sum}}(\boldsymbol{\theta})$. The diagonal joint empirical FIM follows as

$$\text{diag}(\boldsymbol{F}_{\text{joint}}^{\text{emp}}(\boldsymbol{\theta})) = \boldsymbol{g}^2 = \left( \sum_{n=1}^{N} \boldsymbol{g}_n \right)^2.$$

*The key properties of the joint Fisher matrices in Equations* (9) *and* (10) *is that they are squares of sums, and not sums of squares. This provides a theoretical motivation that the square of aggregated gradients indeed corresponds to the diagonal of a Fisher information matrix. In fact, Lin et al. (2024) show that the joint and standard Fisher coincide, $\boldsymbol{F}_{joint}(\boldsymbol{\theta}) = \boldsymbol{F}_{std}(\boldsymbol{\theta})$, and hence that both views— standard and joint—lead to the same underlying Fisher.*

**Handling mini-batching** Note that the joint Fisher's distribution considers a vector of random variables whose size equals the data set. When using only a subset of data, say a mini-batch $\boldsymbol{X}_B = (\boldsymbol{x}_1, \ldots, \boldsymbol{x}_B)$, we can consider the mini-batch version of the joint Fisher, $\boldsymbol{F}_{\text{joint}}(\boldsymbol{\theta}, B)$, which is defined in terms of the marginal distribution $p(\mathbf{y}_B \mid \boldsymbol{X}_B, \boldsymbol{\theta})$ with $\mathbf{y}_B = (y_1, \ldots, y_B)$. Lin et al. (2024) show that $^{N}/_{B} \boldsymbol{F}_{\text{joint}}(\boldsymbol{\theta}, B)$ is an unbiased estimation of $\boldsymbol{F}_{\text{joint}}(\boldsymbol{\theta})$. This property allows us to estimate the Fisher on a larger data set (as would be done when using the standard Fisher in applications) from gradients evaluated on a smaller amount of data (as would be done by an optimizer based on mini-batch gradients). Importantly, this estimation is *unbiased* if we sample labels from the model's likelihood. This ensures that mini-batching itself does not introduce bias in Fisher estimation. However, when labels are replaced with their empirical counterparts (e.g., ground truth labels), the resulting estimator becomes biased. Hence we can think of the standard and joint empirical Fishers as two different biased

approximations of the same underlying Fisher. When using batches, we can simply replace $N$ by $B$ in all expressions that follow.

**Handling averaged loss functions** So far, we have assumed an unscaled loss (1) in our discussion of Fishers. However, most implementations use an average loss

$$\mathcal{L}(\boldsymbol{\theta}) \coloneqq \frac{1}{N} \sum_{n=1}^{N} \ell(f(\boldsymbol{x}_n, \boldsymbol{\theta}), y_n) = \frac{1}{N} \mathcal{L}_{\text{sum}}(\boldsymbol{\theta}). \qquad (11)$$

Note that we cannot absorb the factor $^1/_N$ into the loss function $\ell$ and define $\ell_{\text{scaled}} = ^1/_N \ell$ to reduce Equation (11) to the form of Equation (1), because only $\ell$, but not $\ell_{\text{scaled}}$, corresponds to a probability density via Equation (2). Therefore, we will keep the $^1/_N$ factor separate and use the RHS of Equation (11), which means we can re-use the standard Fisher matrices that are defined in terms of the unscaled loss $\mathcal{L}_{\text{sum}}$ and scale them by $^1/_N$. Rescaling the joint empirical Fisher matrix from Equation (10), we get

$$\frac{1}{N} \boldsymbol{g} \boldsymbol{g}^\top = \frac{1}{N} N \nabla_{\boldsymbol{\theta}} \mathcal{L}(\boldsymbol{\theta}) \left( N \nabla_{\boldsymbol{\theta}} \mathcal{L}(\boldsymbol{\theta}) \right)^\top$$
$$= N \nabla_{\boldsymbol{\theta}} \mathcal{L}(\boldsymbol{\theta}) (\nabla_{\boldsymbol{\theta}} \mathcal{L}(\boldsymbol{\theta}))^\top.$$

In the diagonal Fisher case, we have

$$N \nabla_{\boldsymbol{\theta}} \mathcal{L}(\boldsymbol{\theta})^2 = N \left( \frac{1}{N} \sum_{n=1}^{N} \boldsymbol{g}_n \right)^2. \qquad (12)$$

In practise, re-scaling is often unnecessary as many applications are invariant under scaling the Fisher (Section 3).

**Recycling the squared gradient accumulator** The term in the parenthesis on the RHS of Equation (12) is exactly the quantity whose EMA is computed by the squared gradient accumulator of Equation (8). This leads us to a clear path for using the Squisher, i.e. the **squ**ared gradient accumulator, as an approximation of the F**isher**. Specifically, compared to the Fisher (i.e. the diagonal of the empirical FIM, as commonly used as a measure of parameter importance), the Squisher squares the average gradient over a training batch rather than computing the sum of gradients over an arbitrary collection of datapoints. The Squisher therefore more closely relates to the diagonal of the *joint* empirical FIM. In addition, the squaring of the average gradient introduces a factor of $N$ difference in scale as in Equation (12).

Separately, the Squisher is computed using an EMA of mini-batch gradients. The EMA coefficient $\alpha$ is typically tuned to improve training convergence and therefore might not reflect the best value for approximating the Fisher. For example, the default value of $\alpha$ in Adam (where it is referred to as $\beta_2$) is 0.999, which results in $\boldsymbol{v}^{(t)}$ containing nontrivial contributions from a long history of gradients – the time

constant, i.e. the number of steps to reach a rescaling of $1 - \frac{1}{e} \approx 63\%$, is about 10,000 steps. Averaging over such a long history both introduces contributions from gradients computed with respect to "old" parameter values and also results in a biased estimate of the corresponding joint empirical Fisher computed over all of the data the model has been trained on (Lin et al., 2024).

## 3. Experiments

The above discussion reveals a clear way to relate the squared gradient accumulator to a Fisher, but this relation involves various nontrivial approximations. We therefore turn to exploring whether these approximations are problematic through an empirical study covering a wide range of six settings where the Fisher is used, which we outline below. We emphasize that our goal is not to show that either the Fisher or the Squisher is "better" across all of these settings, but rather to test whether or not the approximations made in formulating the Squisher result in differences in performance compared to using the Fisher itself. To ensure reliable results, we base all of our experiments on prior implementations. In most cases, these implementations used either Adam or AdamW for optimization and therefore lent themselves straightforwardly to using the Squisher. We additionally always compare to a "Fisher-free" baseline, i.e. a method that does not involve computing the Fisher and therefore no additional computational costs (like the Squisher). Our high-level results are shown in Figure 2; complete fine-grained results are provided in Appendix A.1.

> **Naming:** As the empirical diagonal standard Fisher from Equation (7) is the standard choice as a notion of parameter importance in many applications (Matena & Raffel, 2022; Theis et al., 2018; Sung et al., 2021; Achille et al., 2019), we will drop all prefixes and simply refer to it as the "Fisher" from now on.

### 3.1. Fisher Merging

Model merging aims to cheaply combine individual models into a single model that inherits their capabilities (Utans, 1996; Singh & Jaggi, 2020). Fisher merging (Matena & Raffel, 2022) formulates merging as maximizing the joint likelihood of the individual models' posterior distributions over parameters. To do so, Fisher merging uses the Laplace approximation (MacKay, 2003), where the Fisher is the precision matrix of a Gaussian approximation to this posterior. Fisher Merging then uses a closed-form solution to the likelihood maximizing problem:

$$\hat{\boldsymbol{\theta}} = \left( \sum_{i=1}^{M} \boldsymbol{F}_i \right)^{-1} \left( \sum_{i=1}^{M} \boldsymbol{F}_i \boldsymbol{\theta}_i \right) \tag{13}$$

where $\boldsymbol{F}_i$ and $\boldsymbol{\theta}_i$ are the Fisher and parameters of model $i$ out of $M$ models being merged. Equation (13) corresponds to parameter averaging where parameters with higher corresponding values in the Fisher are given a higher weight when averaging. Noting that Equation (13) is invariant to rescaling all $\boldsymbol{F}_i$ by the same constant, using either the Squisher ($N \boldsymbol{v}^{(t)}$) or the squared gradient accumulator ($\boldsymbol{v}^{(t)}$) as Fisher proxies yields the same merge.

**Setup** We directly follow Tam et al. (2024) and merge eight variants of T5-Large-LM-Adapt (Raffel et al., 2020; Lester et al., 2021) that were fine-tuned on text datasets that have been shown to produce performant multitask models (Zhou et al., 2022). Performance is measured as the average accuracy of the merged model on held-out data from the eight datasets used to fine-tune the individual models. As Fisher-free baseline, we use simple parameter averaging (Utans, 1996; Wortsman et al., 2022). For further experimental details, please see Tam et al. (2024, Section 6.2).

**Results** Average performance of the Fisher, Squisher, and unweighted parameter averaging are shown in Figure 2. Overall, we found Squisher merging performed considerably better than Fisher merging. We don't interpret this to mean that the Squisher is "better", but rather that it is due to the inherent instability in multitask merging (Tam et al., 2024; Yadav et al., 2024; Ilharco et al., 2022). Since both the Fisher and Squisher work significantly better than the Fisher-free baseline, we conclude that both provide a reliable estimate of parameter importance. Additionally, we found that Squisher merging performance could suffer if the fine-tuned models were not fully trained, likely because the squared gradient accumulator had not observed sufficient training steps. We explore this factor further in Appendix A.2.

### 3.2. Model Merging by Uncertainty-Based Gradient Matching (UBGM)

Daheim et al. (2024) uncover that "gradient mismatch" arises when merging models that fall in disparate regions of the loss landscape, potentially leading merged models to fall in high-loss areas, thereby degrading merging performance. Daheim et al. (2024) therefore aim to mitigate gradient mismatch by aligning parameter updates with the optimization trajectories of the individual models. Specifically, given a base model with weights $\boldsymbol{\theta}_0$ and Fisher $\boldsymbol{F}_0$, and $M$ fine-tuned models with weights $\boldsymbol{\theta}_i$ and Fishers $\boldsymbol{F}_i$ for $i \in \{1, \dots, M\}$, the parameters of the merged model $\hat{\boldsymbol{\theta}}$ are given by

$$\hat{\boldsymbol{\theta}} = \boldsymbol{\theta}_0 + \left( \boldsymbol{F}_0 + \sum_{i'=1}^{M} \boldsymbol{F}_{i'} \right)^{-1} \left( \sum_{i=1}^{M} (\boldsymbol{F}_0 + \boldsymbol{F}_i)(\boldsymbol{\theta}_i - \boldsymbol{\theta}_0) \right).$$

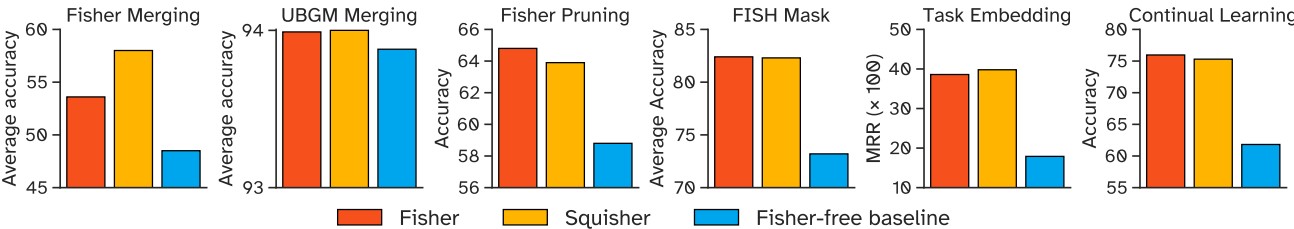

*Figure 2.* Performance of the Fisher, our proposed Squisher (i.e. using the squared gradient accumulator in place of the Fisher), and an applicable Fisher-free baseline across all of the settings we consider. Across all settings, the Squisher performs comparably to the Fisher and outperforms the Fisher-free baseline.

As with Fisher merging, the Squisher simply replaces $\boldsymbol{F}_i$ with the corresponding model's squared gradient accumulator at the end of training.

**Setup**  We exactly replicate the setup of Daheim et al. (2024) and consider the experiments focused on merging fine-tuned variants of RoBERTa (Liu et al., 2019) on four standard text classification tasks. As for Fisher merging, we measure performance in terms of average accuracy on the individual tasks. Daheim et al. (2024) use AdamW for fine-tuning, and we re-use their squared gradient accumulator without modification. Further experimental details are provided in Daheim et al. (2024, section 4.3). Like the previous merging setting, we use parameter averaging as baseline.

**Results**  High-level results are shown in Figure 2. In the UBGM Merging setting, we found that the Squisher performed similarly to Fisher. Interestingly, parameter averaging is a relatively strong Fisher-free baseline in this setup, though both Fisher- and Squisher-based UBGM merging outperform it slightly. Therefore, we consider their performance to be comparable in both settings.

### 3.3. Fisher Pruning

Pruning aims to convert a standard neural network into a "sparse" network where most of the weights are zero (LeCun et al., 1989; Hassibi & Stork, 1992). Theis et al. (2018) formulate pruning as eliminating the parameters that least impact the loss. Given a neural net trained to convergence at $\boldsymbol{\theta}^\star$, the optimal perturbation $\tilde{\boldsymbol{\theta}} = \boldsymbol{\theta}^\star + \boldsymbol{\delta}(i)$, such that $[\tilde{\boldsymbol{\theta}}]_i = 0$ with minimal increase ($\rho$) of the loss is given by

$$\boldsymbol{\delta}(i) = \frac{-[\boldsymbol{\theta}]_i \boldsymbol{F}(\boldsymbol{\theta}^\star)^{-1} \boldsymbol{e}_i}{[\boldsymbol{F}(\boldsymbol{\theta}^\star)^{-1}]_{i,i}}, \quad \rho(\boldsymbol{\delta}(i)) = \frac{[\boldsymbol{\theta}^\star]_i^2}{2[\boldsymbol{F}(\boldsymbol{\theta}^\star)^{-1}]_{i,i}},$$

with $\boldsymbol{e}_i$ the canonical $i$-th basis vector. Assuming a diagonal Fisher produces the pruning statistics $\rho(\boldsymbol{\delta}(i)) = [\boldsymbol{\theta}^\star]_i^2[\boldsymbol{F}(\boldsymbol{\theta}^\star)]_{i,i}/2$. Theis et al. (2018) then retain the parameters corresponding to the $k$ leading pruning statistics.

**Setup**  We base our Fisher pruning experiments on the re-implementation of Lubana et al. (2020). Specifically, we

focus on an experiment described in Section 5.3 of Lubana et al. (2020), which involves pruning a VGG-13 (Simonyan, 2014) network trained on CIFAR-100 (Krizhevsky et al., 2009). Lubana et al. (2020) originally used vanilla stochastic gradient descent for training; we therefore modified the implementation to use Adam and tuned hyperparameters to ensure comparable results. Since Fisher pruning retains the top-$k$ parameters, it is insensitive to global rescaling and we can therefore use the squared gradient accumulator as-is for the Squisher. As a Fisher-free baseline, we consider pruning parameters at random.

**Results**  We present the results for pruning 75% of the model parameters (i.e. reducing the model to 25% of its original size) in Figure 2 and include additional results for 25% and 50% in Appendix A.1. Across all pruning levels, we find that Squisher pruning slightly underperforms Fisher pruning but performs much better than baseline pruning with a random mask.

### 3.4. FISH Mask

Sung et al. (2021) consider sparse training and fine-tuning, i.e. updating only a small subset of a model's parameters during training. They propose FISH Mask, which uses the Fisher to choose which parameters to update during training. Specifically, the $k$ parameters to update are selected based on their importance measured by the Fisher diagonal:

$$\{[\boldsymbol{\theta}]_i \mid [\boldsymbol{F}(\boldsymbol{\theta})]_{i,i} \geq \text{sort}(\text{diag}(\boldsymbol{F}(\boldsymbol{\theta})))_k\}.$$

With the Squisher, we instead update only those parameters with the $k$ largest values in the squared gradient accumulator. As with Fisher pruning, we use random masking as Fisher-free baseline.

**Setup**  We focus on the BERT-Large (Devlin et al., 2019) fine-tuning setting described in Section 4.1 of Sung et al. (2021), which uses the FISH mask to reset parameters of fine-tuned models back to the pre-trained values (i.e. their values before fine-tuning). Specifically, 50% of the model's weights are masked, and the remaining 50% are reset to their pre-trained values rather than retaining their

fine-tuned values. Fine-tuned, masked models are separately trained and evaluated on nine datasets from the GLUE benchmark (Wang et al., 2018). Full details are available in Sung et al. (2021).

**Results** As shown in Figure 2, the Fisher and Squisher attain extremely similar performance when used in the FISH Mask setting. Combined with our previous finding for pruning, this result suggests that the squared gradient accumulator provides a reliable way to rank parameter importance.

### 3.5. Task Embedding

The Task2Vec embedding represents tasks as points in a vector space in which the distance aims to capture task similarity (Achille et al., 2019). Task2Vec embeddings are computed using the Fisher of a model trained on a given task and averaging the values across parameter "groups" (e.g. weight matrices). This vector approximates how sensitive different parameters are to some particular task. Task2Vec has been shown to be useful for predicting task similarities, such as semantic or taxonomic relations, and for determining which tasks are best suited for knowledge transfer (Vu et al., 2020), i.e. whether training on task $a$ before training on task $b$ can improve performance on task $b$. Specifically, the cosine similarity between task vectors $\boldsymbol{F}_a$ and $\boldsymbol{F}_b$ is

$$d_{\text{sim}}(\boldsymbol{F}_a, \boldsymbol{F}_b) = d_{\cos}\left((\boldsymbol{F}_a + \boldsymbol{F}_b)^{-1}\boldsymbol{F}_a, (\boldsymbol{F}_a + \boldsymbol{F}_b)^{-1}\boldsymbol{F}_b\right)$$

and is used to predict task transferability, i.e. tasks with more similar task vectors are predicted as being more amenable to knowledge transfer. Since cosine distance is invariant to rescaling, to use the Fisher we simply replace $\boldsymbol{F}_a$ and $\boldsymbol{F}_b$ with the squared gradient accumulators from the corresponding models.

**Setup** We consider the setting from Vu et al. (2020), who use Task2Vec to predict task transferability for intermediate-task training (Phang et al., 2018; Pruksachatkun et al., 2020). Specifically, Vu et al. (2020) experiment with predicting the best intermediate task for fine-tuning BERT (Devlin et al., 2019) on a wide range of datasets. We focus on the 22 classification, regression, and question-answering datasets. Since the original fine-tuned models were not released by Vu et al. (2020), we re-fine-tuned BERT on each dataset using the AdamW optimizer. As a Fisher-free baseline, we use the simple and effective heuristic of ranking datasets in terms of their size (because larger datasets tend to be more beneficial for intermediate-task transfer (Vu et al., 2020)). We evaluate each ranking method based on its mean reciprocal rank (Voorhees et al., 1999) which measures how a given embedding method tends to rank the best dataset for intermediate-task transfer.

**Results** As seen in Figure 2, the Squisher-based task embedding produced a better mean reciprocal rank than the Fisher-based one; both outperformed the dataset size heuristic. This trend held across both classification/regression and question-answering datasets. This confirms that the squared gradient accumulator's values can be used as a reliable measure of task similarity.

### 3.6. Elastic Weight Consolidation

Continual learning faces the challenge of catastrophic forgetting, where artificial neural networks forget previously learned tasks when training on new tasks. Elastic Weight Consolidation (EWC, Kirkpatrick et al., 2017) aims to mitigate catastrophic forgetting by using the Fisher to avoid changes to model parameters that have a high influence on the performance of previously seen tasks. Specifically, EWC introduces a regularization term that rescales the squared difference between the current parameter value $\boldsymbol{\theta}$ and the learned values from the previous task(s) $\hat{\boldsymbol{\theta}}$ by the Fisher $\boldsymbol{F}$ from the previous task:

$$\mathcal{L}_{\text{EWC}}(\boldsymbol{\theta}) = \frac{\lambda}{2}(\boldsymbol{\theta} - \hat{\boldsymbol{\theta}})^\top \boldsymbol{F}(\boldsymbol{\theta} - \hat{\boldsymbol{\theta}}). \tag{14}$$

**Setup** We focus on task-incremental learning for this study, since EWC has been shown to have poor performance on domain- and class-incremental learning (van de Ven & Tolias, 2019). Task-incremental learning happens when the context identity is known during training, and the model must incrementally learn a set of distinct tasks (Ruvolo & Eaton, 2013). We consider three standard benchmarks for this setting: split MNIST, a split of the original MNIST dataset into five contexts with two digits each (Shin et al., 2017); permuted MNIST, an additional variant of MNIST transformed by applying a fixed, random pixel permutation to each task (Zenke et al., 2017); and split CIFAR100, similarly split into ten contexts with ten classes each (Krizhevsky, 2009). We use an MLP with 478,410 parameters for split MNIST, a larger MLP with 2,126,100 parameters for Permuted MNIST, and a 5-layer CNN with 393,088 parameters for split CIFAR-100. For each protocol, we replaced the Fisher with the Squisher in the EWC regularizer.

Unlike in previous settings, rescaling the Squisher *does* change the learning behavior in this setting as it modifies the regularization strength. For EWC, we found that scaling the Squisher computed on batches of size $B$ by $N$ provided best performance, where $N$ is the data set size on which the original Fisher was computed. Although we lack a formal theoretical justification, this serves as a useful heuristic, i.e., one can start with setting $\lambda_{\text{Squisher}} = N\lambda_{\text{Fisher}}$, and sweep over parameters around this value in a grid search. In practice, when training EWC from scratch, $\lambda_{\text{Fisher}}$ is unknown and must be tuned regardless, so using the Squisher does

not introduce any additional tuning burden. We discuss the importance of adjusting the scaling in Section 4. As a Fisher-free baseline, we incrementally train the model without regularization.

**Results** When used in EWC for continual learning, we find that the Squisher performs slightly better than the Fisher across all three continual learning setups (Figure 2 shows results for CIFAR100; results for other protocols are available in Appendix A.1). Using either the Squisher or the Fisher worked significantly better than using no parameter-wise rescaling (i.e. the identity matrix instead of the Fisher in Equation (14)), suggesting again that the Squisher does capture a reliable notion of parameter importance.

### 3.7. Summary

Across all of the diverse experimental settings we consider, replacing the Fisher with the Squisher had little impact on performance. While performance degraded marginally in Fisher pruning, it improved in Fisher merging, and was almost identical for the other settings.

In all cases, both the Fisher and Squisher significantly outperformed relevant Fisher-free baselines. This validates that the approximations made when going from the Fisher to the Squisher do not significantly impact performance – at least when using diagonal approximations, as is common practise. More broadly, we can confirm that the Squisher meaningfully reflects the importance of a model's parameters to a similar extent to the Fisher.

We want to emphasize that our goal is not to claim that Squisher is inherently superior or inferior to Fisher. The performance differences are problem-dependent and influenced by training trajectories, but overall remain small relative to baseline performance. These differences are largely attributable to noise within the setting, rather than any fundamental distinction between the two approaches. This is further explained in Appendix A.2.

### 3.8. Runtime Analysis

The central motivation of this paper is to obtain *Fishers for free*. The time required to compute the Fisher varies with the model, dataset, and experimental setting; corresponding runtimes are reported in Table 1. In contrast, computing the Squisher incurs virtually no cost, as it simply involves loading pre-computed values and, in the case of EWC, applying a scaling factor. Across all settings, Squisher's runtime remains below 0.1 seconds, and under 1 second for EWC. A detailed breakdown of computation costs across datasets is provided in Appendix A.3.

*Table 1.* Time (in seconds) it took to calculate the Fisher for the settings considered. The Squisher is for free, up to the cost of loading the pre-computed optimizer statistics from disk.

| SETTING | TIME (S) |
|---|---|
| FISHER MERGING | $2.93 \cdot 10^4$ |
| UBGM | $4.36 \cdot 10^2$ |
| FISHER PRUNING | $2.52$ |
| FISH MASK | $6.98 \cdot 10^2$ |
| TASK EMBEDDING | $5.19 \cdot 10^4$ |
| EWC | $1.25 \cdot 10^3$ |

## 4. Ablation Experiments

When relating the squared gradient accumulator to the Fisher in Section 2.3, we highlighted various approximations made. Although our results from Section 3 confirm that these approximations generally do not harm performance, we would still like to untangle each of their impacts.

To do so, we run a series of ablation experiments in the continual learning setting from Section 3.6. We chose continual learning as it was the only setting where results were dependent on appropriately rescaling the Squisher.[2]

**Squared gradient accumulator without rescaling** To measure the importance of rescaling the squared gradient accumulator (as described in Section 2.3), we measure performance when using the squared gradient accumulator directly without tuning the $\lambda$ term in Equation (14). As can be seen in Table 2, not tuning the value and using the default from Fisher provides suboptimal results.

**Changing the exponential moving average** From the results of Fisher merging, we observed that Squisher's performance could degrade if the fine-tuned models were not fully trained, likely because the squared gradient accumulator had not been exposed to enough training steps. We address this effect by reducing $\beta_2$ in Adam, which decreases the emphasis on past squared gradients and places more weight on recent gradients. The optimal setting aligns with the default AdamW value of 0.999, while performance degradation was observed at 0.95. Importantly, our intent is not to tune $\beta_2$ for Squisher; rather, we examine whether any standard optimizer hyperparameters can serve as drop-in approximations for the Fisher. We recommend using the $\beta_2$ that best supports optimization. Notably, even with unusually low values of $\beta_2$, the Squisher continued to outperform the baseline, highlighting that Squisher remains effective as long as conventional training settings are used.

---

[2]Note that some of the ablation conclusions do not hold for split MNIST, likely due to the simplicity of the setting.

*Table 2.* Results from the ablation experiment settings, averaged over 5 trials.

| | FISHER | SQUISHER | SQUISHER W/O NORM. | $\beta_2 = 0.95$ | JOINT |
|---|---|---|---|---|---|
| SPLITM | 99.59 | 98.59 | 98.85 | 98.66 | 99.46 |
| PERMM | 94.55 | 94.47 | 92.05 | 93.37 | 95.59 |
| SPLITC | 75.96 | 75.30 | 61.53 | 72.80 | 75.38 |

**Diagonal joint empirical FIM**  The squared gradient accumulator uses a moving average of gradients over the course of training, whereas FIMs compute an explicit sum of gradients at the end of training. To measure the importance of this difference, we explicitly square the sum of per-example gradients (i.e. we turn off the moving average in the Squisher), shown in Table 2 as JOINT. We see that its performance values are close to FISHER, suggesting the joint Fisher is a useful approximation.

## 5. Related Work

To the best of our knowledge, there has been no prior work investigating whether the squared gradient accumulator can reliably be used as a measure of parameter importance in place of the Fisher. However, there has been ongoing research on more directly connecting adaptive optimizers to second-order optimization. One such example is IVON (Shen et al., 2024), an extension of Adam that approximates the Hessian via weight perturbations. Notably, similar to our goal, IVON demonstrates that its accumulated statistics can serve as a substitute for the FIM in model merging tasks. However, unlike this work, using IVON's statistics involve a nontrivial deviation from what is currently standard practice for training neural networks.

Similarly, the AdaFisher optimizer (Gomes et al., 2024) replaces Adam's second-moment estimation with a novel block-diagonal approximation of the FIM. This approach yields improved performance, emphasizing the richness of the FIM compared to the squared gradient moving average. As with IVON and the Squisher, recycling the statistics produced by AdaFisher or FAdam could yield a similarly effective replacement for the Fisher.

FAdam is another optimizer that leverages the empirical FIM (Hwang, 2024). The authors reinterpret the second-order moment in Adam through the lens of the diagonal Fisher, and propose a modified optimizer to address its limitations. While their work offers valuable insights into the relationship between natural gradient methods and adaptive optimization, it lacks a full theoretical proof and comprehensive empirical analysis.

BackPACK (Dangel et al., 2020) is a library that enables effi-

cient computation of per-sample gradients and second-order quantities like the empirical Fisher diagonal by extending the backward pass of neural networks. While more efficient than naïve for-loop approaches, it still introduces additional overhead, requires code changes, and lacks full architectural support. For instance, it does not support layer normalization. As a result, the for-loop approach remains common in Fisher-based methods. In contrast, the Squisher introduces no extra cost or code modifications, as it reuses the squared gradient accumulator already present in optimizers like Adam (Kingma & Ba, 2014), making it universally supported without these limitations.

## 6. Conclusion

In this paper, we present a novel technique, the Squisher, that serves as a free approximation for the diagonal Fisher in various Fisher-based methods. The Squisher uses gradient accumulators that are readily available during training to provide an estimate of the Fisher, thereby alleviating the costs associated with calculating the Fisher. We motivate this by formulating the empirical Fisher as a joint Fisher, and hence show its parallels with gradient accumulators. We implement the Squisher as a direct replacement for the Fisher in settings where only the relative importance of parameters matters, and apply a rescaling term in contexts where the magnitude of values is consequential. We show that the Squisher had comparable performance to the Fisher across five settings, and had significant improvements from the baselines. We further discuss the impact of the approximations we made by considering several variants.

In future work, we can explore advanced optimizers which use gradient accumulators that approximate the non-diagonal Fisher, and whether that leads to better performance. Another direction is to maintain a moving average of the Fisher over time, analogous to the exponential moving average used in adaptive optimizers, rather than computing it at a single post-training point. Although impractical during real training, it may be interesting to explore whether it offers better approximations for the true Fisher. By reducing the computational burden of Fisher-based methods, this work advances the democratization of deep learning research. Furthermore, it is currently not common practice to share optimizer weights. However, we hope this will encourage the community to share full training configurations, hyperparameters, and optimizer state dicts, fostering more reproducible and inclusive machine learning research.

## Acknowledgements

Resources used in preparing this research were provided, in part, by the Province of Ontario, the Government of Canada through CIFAR, and companies sponsoring the Vector Insti-

tute. We would like to thank Reviewer WT93 for helping us formalize the equivalence between the standard and joint Fisher. This clarification strengthens our motivation to interpret the standard empirical Fisher and Squisher as two distinct empirical approximations of the same underlying Fisher information.

## Impact Statement

This paper presents work whose goal is to advance the field of Machine Learning. There are many potential societal consequences of our work, none which we feel must be specifically highlighted here.

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

# A. Appendix

## A.1. Full Results

We show the full results of all six settings as described in Section 3.

**Fisher Merging**

*Table 3.* FISHER and SQUISHER represent the corresponding performance of the original and proposed method. LINEAR is the unweighted averaging baseline.

| DATASET | FISHER | SQUISHER | LINEAR |
|---|---|---|---|
| COSMOS_QA | 61.0 | 50.7 | 39.4 |
| PAWS | 50.7 | 50.4 | 48.0 |
| QASC | 73.7 | 85.9 | 77.1 |
| QUAIL | 57.2 | 60.7 | 43.2 |
| QUARTZ | 54.6 | 61.1 | 59.3 |
| ROPES | 12.6 | 36.1 | 45.1 |
| SOCIAL_IQA | 68.7 | 61.0 | 48.0 |
| WIKI_QA | 50.1 | 58.1 | 27.6 |
| AVERAGE | 53.6 | 58.0 | 48.5 |

**Model Merging by Uncertainty-Based Gradient Matching**

*Table 4.* FISHER refers to the accuracies using the original method, while SQUISHER represents the accuracies achieved with our proposed estimation technique, and LINEAR is the unweighted averaging baseline. AVERAGE is the direct average across datasets, while TRUE AVERAGE takes into account the number of data samples per dataset.

| DATASET | FISHER | SQUISHER | LINEAR |
|---|---|---|---|
| IMDB | 93.67 | 94.48 | 94.32 |
| YELP | 97.05 | 97.23 | 96.73 |
| RT | 89.68 | 89.02 | 89.02 |
| SST2 | 93.58 | 92.78 | 93.35 |
| AMAZON | 95.97 | 96.47 | 95.97 |
| AVERAGE | 93.99 | 94.00 | 93.88 |
| TRUE AVERAGE | 95.91 | 96.40 | 95.92 |

**Fisher Pruning**

*Table 5.* Pruning is applied to achieve three different target percentages, with each prune percentage–method pair averaged over five seeds. FISHER, SQUISHER, and RANDOM represent the accuracy of their respective pruning methods.

| METHOD | FISHER | SQUISHER | RANDOM |
|---|---|---|---|
| 25% | 67.0 | 66.7 | 65.9 |
| 50% | 66.9 | 65.9 | 64.0 |
| 75% | 64.8 | 63.9 | 58.8 |

## FISH Mask

*Table 6.* GLUE test results with BERT-Large using a FISH mask with a sparsity of 0.50%. FISHER, SQUISHER, and RANDOM represent the accuracy of their respective masking methods.

| TASK | FISHER | SQUISHER | RANDOM |
|---|---|---|---|
| CoLA | 56.5 | 58.8 | 33.5 |
| $MNLI_m$ | 85.2 | 85.5 | 80.0 |
| $MNLI_{mm}$ | 85.7 | 85.8 | 81.3 |
| MRPC | 85.5 | 85.9 | 76.8 |
| QNLI | 93.4 | 93.1 | 84.4 |
| QQP | 87.3 | 87.1 | 83.1 |
| RTE | 67.1 | 65.0 | 60.6 |
| SST-2 | 92.5 | 92.3 | 88.9 |
| STS-B | 88.1 | 87.9 | 70.1 |
| AVG | 82.4 | 82.3 | 73.2 |

**Task Embedding**   We note that because many intermediate datasets produce similar results, the rankings can be noisy. Since we fine-tuned our own models, our rankings did not exactly match those of Vu et al. (2020), but the scores attained were qualitatively similar.

*Table 7.* We report the average rank $\rho$ and NDCG as defined in the original paper, where average rank is the rank assigned to the best source, and NDCG measures the overall ranking quality. We also report Mean Reciprocal Rank (average of the reciprocals of ranks associated with the source) to align with the convention that higher is better, as used in other settings. In-class refers to ranking within the specific task group, and all-class refers to ranking across all datasets, with the number of datasets in each group included in brackets.

| | METRIC | FISHER | SQUISHER | DATASIZE |
|---|---|---|---|---|
| *classification/regression* | | | | |
| IN-CLASS (10) | $\rho$ | 4.6 | 3.8 | 4.8 |
| | NDCG | 82.0 | 84.7 | 81.7 |
| | MRR | 0.421 | 0.468 | 0.136 |
| ALL-CLASS (21) | $\rho$ | 9 | 6.7 | 11.6 |
| | NDCG | 82.5 | 85.9 | 75.0 |
| | MRR | 0.312 | 0.261 | 0.105 |
| *question & answering* | | | | |
| IN-CLASS (10) | $\rho$ | 5.3 | 4.5 | 7.4 |
| | NDCG | 86.3 | 84.0 | 84.4 |
| | MRR | 0.460 | 0.563 | 0.319 |
| ALL-CLASS (21) | $\rho$ | 8 | 5.8 | 10.8 |
| | NDCG | 88.4 | 86.1 | 85.1 |
| | MRR | 0.350 | 0.301 | 0.154 |

**Elastic Weight Consolidation**

*Table 8.* Results from all protocols and scenarios. Note that EWC performs poorly in the class setting (van de Ven & Tolias, 2019). FISHER, SQUISHER, and RANDOM represent the accuracy of their respective continual learning methods.

|  | SCENARIO | FISHER | SQUISHER | BASELINE |
|---|---|---|---|---|
| | TASK | 99.59 | 98.59 | 95.93 |
| SPLIT MNIST | DOMAIN | 65.75 | 69.87 | 56.63 |
| | CLASS | 19.21 | 19.29 | 19.87 |
| | TASK | 94.55 | 94.47 | 78.17 |
| PERMUTED MNIST | DOMAIN | 94.61 | 92.32 | 80.18 |
| | CLASS | 48.25 | 29.64 | 46.73 |
| | TASK | 75.96 | 75.30 | 61.82 |
| SPLIT CIFAR100 | DOMAIN | 18.79 | 20.16 | 16.01 |
| | CLASS | 5.82 | 5.98 | 5.88 |

### A.2. Evaluating the difference in performance between the Fisher and the Squisher

The results in Figure 2 show that it is not consistently the case that either the Fisher or the Squisher performs better. This inconsistency arises due to several layers of approximation. Furthermore, the Squisher relies on a history of gradients, while the Fisher does not. Given that Adam accumulates gradient statistics over time (Kingma & Ba, 2014), it is reasonable to expect that the performance of models using the Squisher depends on how long the model has been trained before extracting the optimizer weights. We illustrate this effect in the two settings with the largest performance gap between the two methods.

**Model Merging**   For Fisher merging, Squisher outperformed Fisher. However, the results reported in this paper are based on the "final model" setting, where all datasets are trained for the same number of epochs without early stopping. An alternative approach, suggested by Tam et al. (Tam et al., 2024), is to merge using the "best model", i.e. merging checkpoints corresponding to the highest individual accuracy.

*Table 9.* Effect of training time for Fisher merging.

|  | FISHER | | SQUISHER | |
|---|---|---|---|---|
|  | FINAL | BEST | FINAL | BEST |
| COSMOS_QA | 61.0 | 62.4 | 50.7 | 31.5 |
| PAWS | 50.7 | 52.0 | 50.4 | 55.8 |
| QASC | 73.7 | 81.1 | 85.9 | 71.7 |
| QUAIL | 57.2 | 55.7 | 60.7 | 41.7 |
| QUARTZ | 54.6 | 59.6 | 61.1 | 56.7 |
| ROPES | 12.6 | 39.8 | 36.1 | 41.9 |
| SOCIAL_IQA | 68.7 | 60.2 | 61.0 | 48.6 |
| WIKI_QA | 50.1 | 61.0 | 58.1 | 92.5 |
| AVERAGE | 53.6 | 59.0 | 58.0 | 55.1 |

The results show that Squisher merging performance can suffer if the fine-tuned models are not fully trained, likely because the squared gradient accumulator had not been computed over sufficiently many training steps. As shown, under the best model setting, the Fisher tends to perform better, but the Squisher performs better when final model is used. This indicates that the merging outcome is sensitive to the specific checkpoints selected, making the comparison between the Squisher and the Fisher unstable. This variability is also reflected in the performance across individual datasets, which differs significantly without a consistent pattern favoring one method over the other.

One possible explanation for the Squisher performing better in the final model setting is that model parameters fluctuate more during early training, making the optimizer-derived Squisher less reliable at that stage. As training progresses and the optimizer statistics stabilize, the Squisher becomes more effective, yielding better merged accuracy.

**Model Pruning**   For this setting, the number of training epochs before pruning was varied, in other words, the number of epochs over which the Adam weights are accumulated. Previously, the model was trained for 15 epochs prior to pruning. Here, we present results for training with only 10 epochs. The results are averaged over 5 runs to illustrate the scale of variance relative to the model's accuracy.

*Table 10.* Effect of training time for Fisher pruning.

| | FISHER | | SQUISHER | |
|---|---|---|---|---|
| EPOCHS TRAINED | 15 | 10 | 15 | 10 |
| ACCURACY | 64.8 | 64.0 | 63.9 | 64.3 |
| STD. DEV. | 0.1 | 0.7 | 0.3 | 0.6 |

Similar to Fisher merging, changing the time trained changes which method performs better. Therefore, this supports the claim that we cannot definitively conclude which method is superior, as the results exhibit considerable variability. But overall, the variation remains small relative to baseline performance and the experimental results thoroughly validate the Squisher as a zero-cost drop-in replacement for the Fisher.

### A.3. Runtime Results

All results are reported in seconds. The Fisher merging and FISH Mask was trained using NVIDIA TITAN Xp, while the other settings were trained on A6000.

**Fisher Merging**

*Table 11.* Time it took to calculate the Fisher for each fine-tuned model to be merged. The individual times were added together to find total runtime cost for Fisher merging.

| DATASET | RUNTIME |
|---|---|
| COSMOS_QA | 4597.096425 |
| SOCIAL_IQA | 6034.254711 |
| PAWS | 8914.329607 |
| QUAIL | 2105.062686 |
| WIKI_QA | 3635.845571 |
| QUARTZ | 522.851436 |
| QASC | 1498.499418 |
| ROPES | 1957.923106 |
| TOTAL | 29265.86296 |

**Model Merging by Uncertainty-Based Gradient Matching**

*Table 12.* Time it took to calculate the Fisher for each fine-tuned model to be merged. The individual times were added together to find total runtime cost for UBGM merging.

| DATASET | RUNTIME |
|---|---|
| IMDB | 88.251802 |
| AMAZON | 298.730116 |
| YELP | 30.893116 |
| ROTTEN TOMATOES | 9.417264 |
| SST2 | 8.584630 |
| TOTAL | 435.876928 |

**Fisher Pruning**

*Table 13.* Time it took to calculate the Fisher, averaged across 5 seeds. A pruning percentage of 75 was used for this runtime comparison, but the timing remains the same across other pruning levels, since the Fisher is computed on the unpruned model, which is identical regardless of the pruning percentage.

| SEED | RUNTIME |
|---|---|
| 0 | 2.53115 |
| 1 | 2.524477 |
| 7 | 2.439246 |
| 42 | 2.578159 |
| 1234 | 2.529666 |
| AVERAGE | 2.5205396 |

**FISH Mask**

*Table 14.* Runtime for computing the Fisher for each task. Since the same model is used across tasks, the runtime remains similar due to consistent model size and computational cost.

| TASK | RUNTIME |
|---|---|
| COLA | 87.106167 |
| MNLI | 86.589986 |
| MRPC | 88.242970 |
| QNLI | 87.799851 |
| QQP | 86.861222 |
| RTE | 87.250311 |
| SST2 | 87.427422 |
| STSB | 86.988816 |
| TOTAL | 698.266745 |

**Task Embedding**

*Table 15.* Time it took to calculate the Fisher for each task. The individual times were added together to find total runtime for this setting.

| DATASET | RUNTIME |
|---|---|
| BOOLQ | 328.932541 |
| COMQA | 2635.093401 |
| CQ | 1150.859747 |
| DROP | 3724.807116 |
| DUORC-P | 6306.916961 |
| DUORC-S | 2546.056582 |
| HOTPOTQA | 8348.362174 |
| NEWSQA | 3798.16761 |
| SQUAD-1 | 920.5654031 |
| SQUAD-2 | 1370.587094 |
| WIKIHOP | 5012.5946521 |
| CoLA | 91.747407 |
| STS-B | 61.246726 |
| WNLI | 7.841009 |
| MNLI | 4030.210978 |
| MRPC | 39.255162 |
| QNLI | 1073.207344 |
| QQP | 3818.912368 |
| RTE | 26.929389 |
| SST-2 | 697.788316 |
| SCITAIL | 245.605103 |
| SNLI | 5666.327469 |
| TOTAL | 51902.01455 |

**Continual Learning**

The time it took to calculate the Fisher after each context is tracked for all three protocols. TOTAL indicates the cumulative time required to compute the Fishers across all contexts.

*Table 16.* Runtime for Split MNIST (in seconds)

| CONTEXT | RUNTIME |
|---|---|
| 1 | 83.543847 |
| 2 | 78.777524 |
| 3 | 83.420858 |
| 4 | 85.942129 |
| TOTAL | 331.684358 |

*Table 17.* Runtime for Permuted MNIST (in seconds)

| CONTEXT | RUNTIME |
|---------|---------------|
| 1 | 1695.310917 |
| 2 | 1699.445545 |
| 3 | 1697.167719 |
| 4 | 1701.106478 |
| 5 | 1689.534109 |
| 6 | 1687.297443 |
| 7 | 1662.508294 |
| 8 | 1686.655048 |
| 9 | 1696.493132 |
| TOTAL | 15215.518690 |

*Table 18.* Runtime for Split CIFAR-100 (in seconds)

| CONTEXT | RUNTIME |
|---------|-------------|
| 1 | 138.255250 |
| 2 | 137.803834 |
| 3 | 137.137637 |
| 4 | 136.980094 |
| 5 | 138.475646 |
| 6 | 139.223644 |
| 7 | 138.512695 |
| 8 | 140.716222 |
| 9 | 139.287938 |
| TOTAL | 1246.392960 |

