# OpenReview forum: "Fishers for Free? Approximating the Fisher Information Matrix by Recycling the Squared Gradient Accumulator"
_ICML.cc/2025/Conference — ICML 2025 spotlightposter_

### Official Review · Reviewer_3M1M · 2025-03-09

**Overall Recommendation:** 2

**Summary:**

- In various contexts where a method is motivated using the Fisher Information Matrix (e.g. EWC, Fisher Pruning, etc), the paper proposes to replace squared sum of gradients, which can sometimes be cumbersome to compute, by the exponential moving average of squared sums of gradients as computed in e.g. Adam, since this quantity is already computed during training. The method is called the "Squisher".
- The proposition is motivated by means of the joint empirical Fisher proposed in Lin et al. 2024
- Empirical validation is performed on a variety of settings that employ approximate diagonal Fisher information, which show that there is no substiantial difference (sometimes slightly better, sometimes slightly worse) in downstream performance when replacing the diagonal second moment of gradient by the Squisher.

**Claims And Evidence:**

The claim is rather conservative (i.e. downstream performance is sometimes better, sometimes not significantly worse) so the empirical results correctly support the claim.

**Essential References Not Discussed:**

For computing per-example gradients (methods):
Goodfellow, I. (2015). Efficient per-example gradient computations. arXiv preprint arXiv:1510.01799.
Rochette, G., Manoel, A., & Tramel, E. W. (2020, November). Efficient Per-Example Gradient Computations in Convolutional Neural Networks. In Workshop on Theory and Practice of Differential Privacy (TPDP).

Closed-form FIM that does not require sampling pseudo target classes:
Pascanu, R., & Bengio, Y. (2013). Revisiting natural gradient for deep networks. arXiv preprint arXiv:1301.3584.

**Experimental Designs Or Analyses:**

Did you check the soundness/validity of any experimental designs or analyses? Please specify which ones, and discuss any issues.

A fundamental flaw in my opinion is that many benchmarked methods claim to be using Fisher Information, but actually end up making at least two stages of approximation:
 - diagonal (whereas there are efficient approximate method nowadays)
 - gradients w.r.t. training set targets, whereas the actual FIM would require gradient w.r.t. sampled targets (there exist closed form).

**Methods And Evaluation Criteria:**

My main concern regarding this paper is that it seems rather limited to only study diagonal approximation of the FIM, and I don't agree with the motivating argument that it is still difficult nowadays to obtain per example gradients e.g. using BackPACK, or other libraries easily found on Github.

Spontaneously, I would have benchmarked full-fledged (i.e. not diagonal) methods using the true FIM, as can be achieved using (E)KFAC approximations e.g. [Ritter et al. 2018] for continual learning.

My feeling after reading the paper (but I might be wrong) is that it is trying to solve a problem which is not anymore so relevant in 2025's deep learning community.

Another concern is the use of the concept of "Fisher" where the Squisher is never actually directly compared to a ground truth diagonal Fisher. From the content of the paper alone, the claimed empirical results could be a side-effect of some desirable property of your new method, that has not much to do with true Fisher information. Even the title of the paper might be misleading.

**Other Comments Or Suggestions:**

Why refer to the FIM's diagonal as the "Fisher" ? If accepted, it might be misleading for future readers, whereas Fisher Information corresponds to the full matrix.

From the body only, please clarify what is new contribution compared to Lin et al.
Moreover, the current paper inherits some ambiguity from Lin et al.'s paper:
 - why motivate a method using an expectation over sampled vectors y, since this expection is never actually computed ?
 - why would it be useful to consider the Riemannian metric defined by the joint empirical FIM ?

**Other Strengths And Weaknesses:**

A main strength of the paper is that many different setups are covered in the experiments.

Another strength is that previous material, as well as the new method, are clearly presented.

**Questions For Authors:**

Why would a data scientist or researcher use the Squisher instead of more accurate approximate methods using github libraries?

**Relation To Broader Scientific Literature:**

The literature regarding diagonal approximate method, applied to various settings (i.e. optimization, pruning, model merging, etc) is correctly discussed. A possible improvement would be to discuss methods that use the actual FIM, and approximate methods that scale its use to large nets.

**Theoretical Claims:**

No new theoretical claims are introduced in the paper as far as I can tell, everything is inherited from Lin et al. 2024.

---

> ### Author Rebuttal · Authors · 2025-04-01
>
> Thanks for your thoughtful feedback, questions, and suggested references.
> We have updated our manuscript in accordance with your suggestions.
> Please find our responses below.
> Let us know if you have any follow-up questions, we'd be happy to discuss further.
>
> ---
>
> > Why would a data scientist or researcher use the Squisher
>
> **Cost:** The main advantage of Squisher is that it’s completely free when using an adaptive optimizer that accumulates the squared gradient (e.g., Adam(W)), which is ubiquitous in deep learning. Since this information is already stored in the optimizer’s state dict, Squisher can be obtained without any additional computation or access to training data. In contrast, computing the Fisher requires both extra computation and access to training data.
>
> **Universal support:** You are right that there exist libraries like BackPACK that compute the empirical Fisher diagonal more efficiently than a for loop.
> However, these alternative approaches still introduce additional cost, require modifications in the code, and do not support all architectures -- for example, BackPACK does not support layer normalization.
> As a result, the for-loop approach remains common in existing Fisher-based methods, including those considered in our paper. Since Squisher simply recycles the already available squared gradient accumulator, it is universally supported without these limitations.
>
> ---
>
> > it seems rather limited to only study diagonal approximation of the FIM
>
> While recent work has explored Fisher approximations beyond the Fisher diagonal, such work has primarily targeted optimization applications (e.g. the natural gradient) as opposed to settings where the Fisher is used as a notion of parameter importance.
> Additionally, such approximations are actually *less* relevant in modern deep learning due to scale.
> Computing, processing, or storing anything larger than the Fisher diagonal (which has as many entries as model parameters) is generally infeasible with today's large-scale models, which might have billions of parameters.
> Consequently, the Fisher diagonal remains especially relevant in today's research.
> Given the increasing popularity of non-diagonal adaptive optimization algorithms like Shampoo/SOAP, we anticipate that their non-diagonal gradient second moments could be used to replace Kronecker-factorized curvature approximations of the Fisher, such as (E)KFAC.
> While we leave this exploration as future work, we believe it actually strengthens our argument for using gradient second moments from adaptive optimizers as Fisher proxies for free.
>
> ---
>
> > the claimed empirical results could be a side-effect of some desirable property of your new method
>
> While we agree that the empirical and true Fisher can meaningfully differ (as pointed out by other works, e.g. https://arxiv.org/pdf/1905.12558), most work using the Fisher as a notion of parameter importance uses the empirical Fisher because it produces good downstream performance and is more straightforward and efficient to compute than the true Fisher.
>
> Please also see our response to the second quoted note in Reviewer WT93.
>
> ---
>
> > Why refer to the FIM's diagonal as the "Fisher" ?
>
> Please see our response to the same question in Reviewer WT93 (third quoted note).
>
> ---
>
> > [...] please clarify what is new contribution compared to Lin et al.
>
> The work of Lin et al. serves as one motivation for our work and provides perspective on Fisher approximations that first sum then square gradients. Our study aims primarily to empirically validate the connetion between the Squisher and Fisher; we do not make or claim any substantial theoretical contributions.
> Note that Lin et al. primarily focus on optimization and the role of the square root in the update of adaptive optimizers for neural network training. This is different from our setups, and their empirical findings do not overlap with ours.
>
> ---
>
> > - why motivate a method using an expectation over sampled vectors y, since this expection is never actually computed ?
> > - why would it be useful to consider the Riemannian metric defined by the joint empirical FIM ?
>
> The motivation for considering the joint Fisher which stems from considering a vector of labels is that it gives rise to a Fisher which first sums, then squares the gradient.
> This pattern is similar to the statistics of adaptive optimizers like Adam that accumulate the square of summed gradients.
>
> As pointed out by Reviewer WT93, the joint and standard Fishers coincide, i.e. they approximate the same underlying "true" Fisher.
> Therefore, the standard empirical Fisher and the joint empirical Fisher can be regarded as two different empirical approximations of the same underlying Fisher.
> Since the standard empirical Fisher is used in various problems in practise, our paper studies if the Squisher, which approximates the joint empirical Fisher, can serve as an a "free" drop-in replacement.

---

### Official Review · Reviewer_D2a9 · 2025-03-12

**Overall Recommendation:** 4

**Summary:**

This paper introduces "Squisher," a method that repurposes the squared gradient accumulator from adaptive optimizers (such as Adam) to approximate the Fisher Information Matrix (FIM) without additional computational cost. The authors provide theoretical analysis connecting the squared gradient accumulator to the FIM and evaluate the approach across six settings:

- Fisher Merging
- Uncertainty-Based Gradient Matching (UBGM)
- Fisher Pruning
- FISH Mask (sparse training)
- Task Embedding with Task2Vec
- Elastic Weight Consolidation (continual learning)

The main finding is that Squisher consistently performs similarly to the traditional Fisher while outperforming Fisher-free baselines, making Fisher-based methods more practical by eliminating their computational overhead.

**Claims And Evidence:**

The claims are well-supported through:

- Theoretical analysis connecting squared gradient accumulators to the Fisher, particularly through the "joint Fisher" formulation
- Comprehensive empirical evaluation across six different applications
- Ablation studies examining the impact of specific approximations (normalization and EMA coefficient)
- Honest reporting of both positive and negative results

The experimental results convincingly demonstrate that Squisher can effectively replace the Fisher in various applications without significant performance degradation.

**Essential References Not Discussed:**

The paper covers most relevant prior work. However, it could benefit from more discussion of recent second-order optimizers:

- Related work: The paper does not mention some other recent second-order optimizers such as AdaFisher [1], which could be relevant in the context of adaptive second-order methods. Including these in the discussion would provide a more comprehensive overview of related work.

[1] Gomes, D.M., Zhang, Y., Belilovsky, E., Wolf, G., & Hosseini, M.S. (2025), AdaFisher: Adaptive second order optimization via fisher information. In the thirtheenth international conference on learning representations.

**Experimental Designs Or Analyses:**

The experimental design is sound:

- The authors base their implementations on prior work for direct comparability
- They use multiple random seeds where appropriate (e.g., five seeds for pruning experiments)
- Both positive and negative results are reported transparently
- The ablation studies in Section 4 isolate specific factors affecting performance

**Methods And Evaluation Criteria:**

The methods and evaluation criteria are appropriate:

- The authors replicate established experimental setups from prior work for each application
- For each setting, they compare against both the original Fisher implementation and a Fisher-free baseline
- The evaluation metrics are standard for each application (accuracy, mean reciprocal rank, etc.)
- Multiple random seeds are used for stochastic elements

**Other Comments Or Suggestions:**

The paper is well-written and clearly structured. No significant typos or errors were noted.

**Other Strengths And Weaknesses:**

Strengths:

- Practical impact: Eliminates the computational overhead of Fisher-based methods
- Broad evaluation: Demonstrates robustness across diverse applications
- Clear theoretical grounding: Well-explained connection to the Fisher
- Honest reporting: Presents both positive and negative results

Weaknesses:

- Limited insight into performance differences: More analysis could explain why Squisher outperforms Fisher in some settings but underperforms in others
- Hyperparameter sensitivity: The ablation study shows dependence on the EMA coefficient, but provides limited guidance on optimal settings
- Some implementation details could be clearer, particularly for mini-batch scaling

**Questions For Authors:**

1- In Fisher Merging experiments, Squisher significantly outperforms the original Fisher. Do you have insights into why this is the case, rather than just performing similarly? Does the Squisher have properties particularly beneficial for model merging?

2- How sensitive is Squisher to optimizer hyperparameters, particularly the EMA coefficient? Given these are typically tuned for optimization rather than Fisher approximation, do you have recommendations for non-optimal settings?

3- How does mini-batch size affect the quality of the Squisher approximation? Since the scaling factor is related to dataset size N, are there special considerations for very large datasets with small mini-batches?

**Relation To Broader Scientific Literature:**

The paper positions its contributions well within multiple research areas:

- Natural Gradient Descent and second-order optimization methods
- Adaptive gradient methods (Adam, RMSProp, etc.)
- Applications of Fisher diagonal as a parameter importance measure
- The practical challenges of computing the Fisher in existing frameworks

The authors correctly identify that computational costs have hindered the adoption of Fisher-based methods, motivating their approach.

**Theoretical Claims:**

I checked the theoretical derivations connecting the squared gradient accumulator to the Fisher Information Matrix. The connections drawn between different formulations (particularly the joint empirical Fisher and its relationship to the accumulator in Equations 11-15) appear sound. The scaling factor of N needed to align the Squisher with the Fisher (Equation 15) is correctly identified and explained.

---

> ### Author Rebuttal · Authors · 2025-04-01
>
> Thanks for your thoughtful feedback and suggestions.
> We will make sure to mention AdaFisher as adaptive method that uses a diagonal Kronecker-factored approximation of the Fisher.
> Please find our responses below and let us know if you have follow-up questions.
>
> ---
>
> > Limited insight into performance differences: More analysis could explain why Squisher outperforms Fisher in some settings but underperforms in others.
>
> > Q1- In Fisher Merging experiments, Squisher significantly outperforms the original Fisher. Do you have insights into why this is the case, rather than just performing similarly? Does the Squisher have properties particularly beneficial for model merging?
>
> Since both the diagonal Fisher and Squisher rely on multiple approximation steps, it is difficult to isolate a specific factor responsible for their divergence. We want to emphasize that our goal is not to claim that Squisher is inherently superior or inferior to Fisher. The performance differences are problem-dependent and influenced by training trajectories, but overall, the variation remains small relative to baseline performance and our experimental results thoroughly validate the Squisher as a zero-cost drop-in replacement for the Fisher.
>
> Let us illustrate this point for two settings with the largest difference:
>
> - For merging, both Fisher and Squisher exhibit high variance in performance depending on the checkpoint used.
>   We will describe these variances in more detail in the text and quantify them with error bars.
> - For pruning, we will add results under varying the training duration.
>   The table below shows that decreasing the number of epochs slightly increases the Squisher's performance.
>
>   | Method | Fisher| Fisher |Squisher | Squisher|
>   | -------- | -------- | -------- | -------- | -------- |
>   | Epochs Trained     | 10     | 15     |10     | 15     |
>   | Accuracy    | 64.0   | 64.8     |64.3     | 63.9     |
>   | Std dev    | 0.7     | 0.1     |0.3     | 0.6     |
>
> ---
>
> > Hyperparameter sensitivity: The ablation study shows dependence on the EMA coefficient, but provides limited guidance on optimal settings
>
> > Q2- How sensitive is Squisher to optimizer hyperparameters, particularly the EMA coefficient? Given these are typically tuned for optimization rather than Fisher approximation, do you have recommendations for non-optimal settings?
>
> The ablation results show the effect of varying the β₂ hyperparameter. We found that lowering β₂ hurt performance, highlighting the importance of accumulating gradient information over time. The optimal setting aligns with the default AdamW value of 0.999, while performance degradation was observed at 0.95. Notably, even with an unconventionally low β₂, the Squisher estimate still outperformed the baseline, indicating that while β₂ impacts performance, it remains usable if conventional values are used during training. We will clarify this in the final report, and we can provide a more comprehensive plot of β₂ values vs accuracy if necessary.
>
> ---
>
> > Some implementation details could be clearer, particularly for mini-batch scaling
>
> Thanks for bringing this to our attention.
> We will make sure to update Figure 1 to incorporate an additional stage where mini-batching is introduced.
> Let us know if this addresses your concern or if there are specific details you’d like us to clarify further.
>
> ---
>
> > Q3- How does mini-batch size affect the quality of the Squisher approximation? Since the scaling factor is related to dataset size N, are there special considerations for very large datasets with small mini-batches?
>
> This is a theoretically interesting question and we will try to add some more experiments on the EWC setting with varying batch sizes.
> Practically though, we believe that varying the batch size should not be considered a hyperparameter of Squisher, since changing it will also impact the optimization algorithm.
> Its choice should primarily be influenced by the optimization algorithm and the hardware.
> Generally speaking, our results did not exhibit a major dependence on batch size, and we anticipate that the Squisher should provide reasonable performance for typical batch sizes.

---

### Official Review · Reviewer_WT93 · 2025-03-14

**Overall Recommendation:** 4

**Summary:**

This paper proposes reusing the squared gradient estimator of adaptive gradient methods as an approximation to the Fisher information matrix, called the 'Squisher'. Through extensive evaluation in model merging, model pruning, sparse fine-tuning, task embedding, and continual learning, the authors demonstrate that the Squisher performs on par with the empirical Fisher without needing additional gradient computations.

## Update after the rebuttal

The authors addressed my remaining concerns. I retain my recommendation of acceptance.

**Claims And Evidence:**

The paper's claims, mostly of a performance-centric nature, are well-supported. In particular, the paper claims that the Squisher approximator is on par with the empirical Fisher on a wide range of tasks, _not_ that it is a good approximation of either the empirical Fisher or even the true Fisher (which would require further theoretical analysis on the quality of the approximations, needing even distributional assumptions).

**Essential References Not Discussed:**

I am unaware of _essential_ references that the paper does not consider.

**Experimental Designs Or Analyses:**

I reviewed the experimental design in detail, in particular, how the diagonal Fisher approximations are used in the downstream tasks and on what datasets and architectures the evaluation happens. This work uses T5 and BERT Transformer models for text-based tasks, and a VGG-13 for Fisher Pruning. I am satisfied with these choices. The exact model used in the EWC experiment is unspecified.

**Methods And Evaluation Criteria:**

The authors evaluate the Squisher against both the empirical Fisher and a Fisher-free baseline. The downstream tasks include model merging and pruning, sparse fine-tuning, and continual learning. These applications all require a notion of parameter importance, making them appropriate downstream tasks. The datasets and architectures are all large-scale.

**Other Comments Or Suggestions:**

- I prefer retaining the term "Fisher" for the Fisher information matrix and keeping the "empirical" adjective for cases when the ground-truth labels are used. These terms are often used interchangeably in the literature, and simplifying the naming in this paper could potentially add to the confusion.
- Eq. (10): $X_n$ should be $X$. The same holds for the Joint Fisher in Fig. 1, except there, the $\frac{1}{N}$ scaling is also incorrect.
- Eq. (8): I suggest using $B$ instead of $N$ to prevent confusion about mini-batching vs. full-batch training.
- Eq. (15): I suggest adding ", where squaring is performed element-wise" after the equation. Likewise in Eqs. (15, 16) for division and multiplication.
- L303: "section 5.3" -> "Section 5.3"
- L293: "section 4.1" -> "Section 4.1"
- L295: "uses <the> FISH mask"
- The first paragraph of Section 3.5 might be hard to understand for readers unfamiliar with Task2Vec. I propose the following clarification: "Task2Vec embeddings are computed using the Fisher of a model trained on a given task. ~~and averaging~~ The values are averaged across parameter ``groups'' (e.g. weight matrices), leading to a smaller-dimensional representation than the number of weights." In the Setup paragraph, consider elaborating on the notion of "intermediate task". It might be instructive to state the full pipeline (pretraining -> intermediate task -> downstream task).
- L330: "dataset into five contexts"

**Other Strengths And Weaknesses:**

A clear strength of this paper is its empirical evaluation. Numerous diverse downstream tasks considered where the Squisher performs on par with the empirical Fisher. Further, the proposed method recycles gradient estimators inherent to adaptive gradient methods, imposing virtually no overhead for calculating the Squisher. This insight, combined with the poor baseline performance on downstream tasks, motivates the proposed method well. Lastly, the paper is enjoyable to read and has a clear structure.

A weakness of this paper is the lack of motivation on why the empirical version of the "new Fisher matrix" of Lin et al. (2024) is a sensible alternative object to describe parameter importance. This is a conceptually simple fix: one can show that the underlying "true" Fisher remains the same and only the nature of the approximation differs. In detail, for $n$ i.i.d. input random variables (RVs) $X$ and output RVs $Y$, distributed according to some parametric process $p(x, y \mid \theta) = p(y \mid x, \theta)p(x)$, the "true" Fisher is written as $$\begin{align}\mathbb{E}\_{p(X, Y \mid \theta)}\left[{\nabla}\_{\theta} \log p(Y \mid X, \theta){\nabla}\_{\theta} \log p(Y \mid X, \theta)^\top\right].\end{align}$$ Using the additivity of the Fisher information for i.i.d. random variables and a discrete set of inputs {$x_1, \dots, x_N$} to approximate $p(x)$, we obtain $$\begin{align}\sum_n \mathbb{E}\_{p(y \mid x_n, \theta)}\left[\nabla\_\theta \log p(y \mid x_n, \theta) \nabla\_\theta \log p(y \mid x_n, \theta)^\top\right].\end{align}$$ The standard empirical Fisher approximation substitutes labels {$y_1, \dots, y_N$} from the true generative model:
$$\begin{align}\sum_n \nabla\_\theta \log p(y_n \mid x_n, \theta) \nabla\_\theta \log p(y_n \mid x_n, \theta)^\top.\end{align}$$
Instead, the formula the authors use can be derived from the "true" Fisher by first Monte Carlo approximating $X$ with a single set of inputs $\mathcal{X} =$ {$x_1, \dots, x_N$}, resulting in:
$$\begin{align}\mathbb{E}\_{p(Y \mid \mathcal{X}, \theta)}\left[{\nabla}\_{\theta} \log p(Y \mid \mathcal{X}, \theta){\nabla}\_{\theta} \log p(Y \mid \mathcal{X}, \theta)^\top\right],\end{align}$$
then substituting in the ground-truth labels $\mathcal{Y} = $ {$y_1, \dots, y_N$}:
$${\nabla}\_{\theta} \log p(\mathcal{Y} \mid \mathcal{X}, \theta){\nabla}\_{\theta} \log p(\mathcal{Y} \mid \mathcal{X}, \theta)^\top.$$

It would also be insightful to see the performance of the true (MC-approximated) Fisher on the downstream task for comparison, as the empirical approximation loses theoretically desirable properties of the Fisher.

**Questions For Authors:**

1. In the paragraph of L292, the authors discuss the fine-tuning setting described in Section 4.1 of Sung et al. (2021). According to this description, it uses the FISH mask to reset parameters. I have not found this in Section 4.1 of Sung et al. (2021); could the authors clarify what they mean by resetting parameters?
2. "Another experiment is to maintain a moving average of Fishers": Do the authors refer to the full Fisher information matrix as the Fisher here instead of the diagonal? Otherwise, Adam already calculates this quantity.

**Relation To Broader Scientific Literature:**

The Fisher Information Matrix, proposed by Fisher (1922), is a natural (degenerate) Riemannian metric on the parameter space of neural networks (Amari, 1998). While Kunstner et al. (2019) argue against the use of the _empirical_ Fisher as a _curvature estimate_, it still gained widespread use as an indicator of _parameter importance_. This work makes use of existing buffers in adaptive gradient methods to provide a cheap approximation to the Fisher. Lin et al. (2024) use this approximation to motivate the removal of the square root in adaptive gradient methods and show that it does not suffer from the problems of the empirical Fisher outlined in (Kunstner et al., 2019). This work uses this "free" quantity in downstream tasks ranging from continual learning to model merging.

**Theoretical Claims:**

No such theoretical claims were made in the submission.

---

> ### Author Rebuttal · Authors · 2025-04-01
>
> Thanks a lot for your thorough review and the various suggestions that we will make sure to incorporate into the manuscript.
>
> We are specifically grateful to the reviewer for pointing out the equivalence between the standard and joint Fisher, which strengthens our motivation to view the standard empirical Fisher and Squisher as two different empirical approximations of the same underlying Fisher.
>
> Please find our response to your remaining concerns below.
> Let us know if you have follow-up questions or suggestions.
>
> ---
>
> > The exact model used in the EWC experiment is unspecified.
>
> We used a 5-layer CNN with 393,088 parameters for the incremental learning task on CIFAR-100.
> We will add details of its architecture to our draft.
> Please let us know if you would like further clarification.
>
> ---
>
> > It would also be insightful to see the performance of the true (MC-approximated) Fisher on the downstream task for comparison, as the empirical approximation loses theoretically desirable properties of the Fisher.
>
> For the EWC setting, we actually computed the true diagonal Fisher for the results presented in the paper. As shown, Squisher provided similar performance to the true diagonal Fisher. We used the true Fisher in this case because it was part of the original implementation.
>
> We agree that these additional experiments would be interesting for other settings. We will explore their feasibility, though some may require a large number of MC samples, making them impractically expensive.
>
> Note that this does not affect the main contributions of our paper, which is proposing a cheap replacement for the empirical Fisher diagonal, which is widely used in the tasks we study, specifically at extreme scales.
>
> ---
>
> > - I prefer retaining the term "Fisher" for the Fisher information matrix and keeping the "empirical" adjective for cases when the ground-truth labels are used. These terms are often used interchangeably in the literature, and simplifying the naming in this paper could potentially add to the confusion.
>
> We agree that, given the frequent confusion of Fisher and empirical Fisher in the community, we should make the terminology less ambiguous.
> We will do so by introducing acronyms for the attributes **diagonal** and **empirical** to specify the Fishers used in the text more clearly.
>
> ---
>
> > Q1. In the paragraph of L292, the authors discuss the fine-tuning setting described in Section 4.1 of Sung et al. (2021). According to this description, it uses the FISH mask to reset parameters. I have not found this in Section 4.1 of Sung et al. (2021); could the authors clarify what they mean by resetting parameters?
>
> For the FISH mask setting, the diagonal Fisher is used to determine which parameters to mask by identifying those with the lowest estimated importance. The remaining parameters are retained and fine-tuned on the new task. By "resetting parameters," we refer to resetting the masked weights to their original pre-trained values rather than using from their fine-tuned values.
>
> We will clarify this in the final paper. Please let us know if you need further details.
>
> ---
>
> > Q2. "Another experiment is to maintain a moving average of Fishers": Do the authors refer to the full Fisher information matrix as the Fisher here instead of the diagonal? Otherwise, Adam already calculates this quantity.
>
> Apologies for the confusion.
>
> Typically, the diagonal Fisher is computed at a single point in time, after training for a specific number of epochs.
> The ablation we propose is maintain a moving average over these values, which is similar to the exponential moving average in adaptive optimizers.
> The key distinction between this approach and what Adam computes lies in the order of operations: Adam maintains a moving average of squared gradients (i.e., sum then square), whereas the Fisher is derived from the squared expectation of gradients (i.e., square then sum), as discussed in the background section.

---

### Official Review · Reviewer_deFR · 2025-03-17

**Overall Recommendation:** 4

**Summary:**

This paper explores the idea of approximating the Fisher Information Matrix by using the squared gradient accumulator that is already computed in optimizers like Adam. The authors have done an excellent job of testing this approximation in six different applications where empirical Fisher is used and show that in most of the settings, this cheap approximation is good enough.

**Claims And Evidence:**

Yes. The authors were very careful in not claiming that the proposed approximation is consistently better than using the computationally more expensive Fisher. There is no overclaiming.

**Essential References Not Discussed:**

No. I appreciate the author's effort in giving credit to Adadelta, while most papers cite RMSProp for the accumulator mechanism.

**Experimental Designs Or Analyses:**

Yes. No issues with the experimental design.

**Methods And Evaluation Criteria:**

Yes.

**Other Comments Or Suggestions:**

* page-2: Adam citation is not correct.

* In all the results sections, while you mention that squisher is better than or on par with Fisher, you should also highlight that it is actually much cheaper to compute squisher. This is implicit but you can highlight this better in your experimental sections.

**Other Strengths And Weaknesses:**

Strength:

* The paper considers a computationally cheap approximation for diagonal Fisher and demonstrates that it works on par or better than the computationally costlier methods.

**Questions For Authors:**

* Q1: page 4: In the last paragraph of section 2, you talk about the effect of having \beta_2 = 0.999 and mention that it results in a biased estimate. But then the discussion is not complete. Is it good or bad that the estimate is biased?

* Q2: For all the applications, the results section just mentions whether squisher underperforms or overperforms Fisher. But you do not reason why in every application. It might be useful to add this discussion.

**Relation To Broader Scientific Literature:**

The authors have done a good job of positioning their work. They took a really simple idea of coming up with a cheap approximation for Fisher but then did an extensive empirical analysis to understand if the approximation was reasonable.

**Theoretical Claims:**

There are no proofs in the paper.

---

> ### Author Rebuttal · Authors · 2025-04-01
>
> Thanks for your effort and valuable feedback.
> We fixed the Adam citation in the manuscript, thanks for pointing that out.
> Please find our responses below and let us know if you have any follow-up questions.
>
> ---
>
> > [...] while you mention that squisher is better than or on par with Fisher, you should also highlight that it is actually much cheaper to compute squisher.
>
> Thanks for this actionable suggestion!
> We will make sure to highlight this more clearly, and also include a small table in the main text that contrasts the computation times of the empirical Fisher with that of the Squisher (essentially zero).
>
> ---
>
> > Q1: page 4: In the last paragraph of section 2, you talk about the effect of having $\beta_2$ = 0.999 and mention that it results in a biased estimate. But then the discussion is not complete. Is it good or bad that the estimate is biased?
>
> Thanks for bringing this to our attention, let us clarify (we will do so in the text, too):
>
> There are two different types of bias in the Squisher.
> The first stems from approximating the Fisher with the empirical distribution.
> The paragraph tries to highlight the second one, which stems from the optimization algorithm's exponential moving average heuristic (using $\beta_2$).
> While the empirical Fisher is computed at one point in parameter space, the Squisher contains contributions from an entire trajectory.
>
> We can only hypothesize if these averages are good or bad (e.g. for optimization they definitely seem to help as the default value for $\beta_2$ in Adam is $0.999$).
> Instead, we accept the presence of exponential moving averages as given, since our goal is to investigate whether the optimizer's statistics can be used as drop-ins for the Fisher.
>
> In our ablation experiments, we found that the Squisher still works reasonably (i.e. better than the Fisher-free base line) even if we vary $\beta_2$, but the default value seems to work best in most cases.
>
> ---
>
> > Q2: For all the applications, the results section just mentions whether squisher underperforms or overperforms Fisher. But you do not reason why in every application. It might be useful to add this discussion.
>
> Please see our response to the same question in Reviewer D2a9 (first quoted note).

---

### Decision · Program_Chairs · 2025-05-01

**Decision:**

Accept (spotlight poster)

**Comment:**

The paper proposes to recycle the squared-gradient accumulator used in common optimizers such as AdamW as an approximation to the Fisher matrix. Across several benchmarks, consistent improvements of the squared-gradient accumulator over the diagonal Fisher are demonstrated.

Overall, the paper makes an important contribution and may put more awareness in the community that variance estimates computed during optimization can be reused. The contribution was highly appreciated by most reviewers.

The main criticisms by the reviewers were:
1) the lack of systematic comparisons whether better Fisher approximations give better downstream performance,
2) extension to non-diagonal Fishers,

I found these to be addressed by the rebuttal, and given today's billion-parameter scale models these are hard to run and properly benchmark.

Overall, I recommend acceptance of this work and believe that it will make a nice contribution to the conference and potentially an impact (e.g., people releasing AdamW's scale vector on HuggingFace more often along with the trained parameters). For the final version, perhaps it could be acknowledged more clearly that the idea of reusing the squared-gradient accumulator for model merging was already mentioned at the end of Section 3.3 in Daheim et al. 2024.